# Function of Porous Carbon Electrode during the Fabrication of Multiporous-Layered-Electrode Perovskite Solar Cells

**Ryuki Tsuji** [1], **Dmitry Bogachuk** [2], **David Martineau** [3], **Lukas Wagner** [2], **Eiji Kobayashi** [4],
**Ryoto Funayama** [4], **Yoshiaki Matsuo** [5], **Simone Mastroianni** [2], **Andreas Hinsch** [2]
**and Seigo Ito** [1,*]

[1]  Department of Materials and Synchrotron Radiation Engineering, Graduate School of Engineering,
    University of Hyogo, 2167 Shosha, Himeji, Hyogo 671-2280, Japan; r2g128@gmail.com
[2]  Fraunhofer Institute for Solar Energy Systems ISE, Heidenhofstraße 2, D-79110 Freiburg, Germany;
    dmitry.bogachuk@ise.fraunhofer.de (D.B.); lukas.wagner@ise.fraunhofer.de (L.W.);
    andreas.hinsch@ise.fraunhofer.de (A.H.)
[3]  Solaronix SA, Rue de l'Ouriette 129, CH-1170 Aubonne, Switzerland; david.martineau@solaronix.com
[4]  Kishu Giken Kogyo Co., Ltd., 446 Nunohiki, Wakayama 641-0015, Japan;
    ekobayashi@kishugiken.co.jp (E.K.); rfunayama@kishugiken.co.jp (R.F.)
[5]  Department of Applied Chemistry, Graduate School of Engineering, University of Hyogo, 2167 Shosha,
    Himeji, Hyogo 671-2280, Japan; ymatsuo@eng.u-hyogo.ac.jp
*   Correspondence: itou@eng.u-hyogo.ac.jp; Tel.: +81-79-267-4150

**Abstract:** We demonstrate the effect of sheet conductivity and infiltration using the example of two graphite types, showing that, in general, the graphite type is very important. Amorphous and pyrolytic graphite were applied to carbon electrodes in fully printable carbon-based multiporous-layered-electrode perovskite solar cells (MPLE-PSCs): <glass/F-doped $SnO_2$/compact-$TiO_2$/porous-$TiO_2$+perovskite/porous-$ZrO_2$+perovskite/porous-carbon+perovskite>. The power conversion efficiency (*PCE*) using amorphous graphite-based carbon (AGC) electrode was only 5.97% due to the low short-circuit photocurrent density ($J_{sc}$) value, which was due to the low incident photon-to-current efficiency (IPCE) in the short wavelength region caused by the poor perovskite filling into the porous $TiO_2$-$ZrO_2$ layers. Conversely, using pyrolytic graphite-based carbon (PGC) electrode, $J_{sc}$, open-circuit photovoltage ($V_{oc}$), fill factors (*FF*), and *PCE* values of 21.09 mA cm$^{-2}$, 0.952 V, 0.670, and 13.45%, respectively, were achieved in the champion device. PGC had poorer wettability and a small specific surface area as compared with AGC, but it had better permeability of the perovskite precursor solution into the porous $TiO_2$/$ZrO_2$ layers, and therefore a denser filling and crystallization of the perovskite within the porous $TiO_2$/$ZrO_2$ layers than AGC. It is confirmed that the permeability of the precursor solution depends on the morphology and structure of the graphite employed in the carbon electrode.

**Keywords:** perovskite solar cells; carbon electrode; graphite; porous electrode; permeability

## 1. Introduction

Organic-inorganic metal halide perovskite solar cells (PSCs) have been improving rapidly, and their power conversion efficiency (*PCE*) has increased from 3.8% in 2009 to 25.2% in 2020 [1,2]. Because PSCs can be produced easily and inexpensively by simple processes such as printing and coating, worldwide research and development activities have been conducted on the commercialization of this technology. However, since the organic substances used in PSCs are unstable with respect to

ambient air, moisture, and heat, recently, there has been a great deal of interest in improving durability. In addition, the components of PSCs, such as hole transport materials (e.g., spiro-OMeTAD) and metal counter electrodes (e.g., Au or Ag), are expensive, and utilization of such materials can be a hindrance for low-cost PSCs. Using only low-cost materials (without Au, Ag, and spiro-OMeTAD), in 2013, Ku, et al. (H. Han's group, HUST, Wuhan, China) first reported hole-conductor-free PSCs with carbon electrodes as counter electrodes instead of expensive metal electrodes [3]. These carbon-based hole conductor-free PSCs consist of scaffold layers (an electron transport layer (ETL, e.g., mesoporous $TiO_2$), a spacer layer (SL, e.g., mesoporous $ZrO_2$ and $Al2O_3$), and a carbon back contact electrode) (Figure 1a,b). All layers can be deposited by screen printing processes. In general, the m-$TiO_2$ layer and m-$ZrO_2$ layer are sintered at 500 °C, and the carbon layer is sintered at 400 °C. The m-$TiO_2$ and m-$ZrO_2$ layer are sintered simultaneously or separately. In this work, the multiporous-layered-electrode perovskite solar cells (MPLE-PSCs) are fabricated by simultaneous sintering, which is a simpler fabricating process. As the final processing step, the perovskite precursor solution is drop-casted on the carbon layer and penetrates into the scaffold layers. The perovskite crystals are formed by annealing and drying the precursor to complete the device manufacturing. One of the challenges of these fullyprintable multiporous-layered-electrode perovskite solar cells (MPLE-PSCs) is the homogeneous formation of perovskite crystals inside the mesoporous layers that are covered by a thick graphite back electrode (totally, more than 10 μm thick). One way to achieve perfect pore filling and higher performance is to apply mixed two-dimensional and three-dimensional (2D/3D) perovskite $(5\text{-AVA})_x(MA)_{1-x}PbI_3$ as light absorbers (5-AVA, 5-aminovaleric acid and MA, methylammonium) [4]. The template effect of 5-aminovaleric acid iodide (5-AVAI) in crystal growth promotes perovskite crystal growth in the pores, enabling the fabrication of high-performance devices. In addition, MPLE-PSCs are known to have higher long-term stability as compared with standard thin-film PSCs [5–7]. This is because of the thick carbon electrode (>10 μm), which is chemically stable and protects the photoactive perovskite crystal in the scaffold layers from moisture and oxygen in the ambient atmosphere. The high durability brought by the inexpensive carbon electrodes of MPLE-PSCs is a key technology for commercialization of PSCs, and it has already been shown to be stable in a small module for about one year [8,9]. Hence, MPLE-PSCs have rapidly been attracting attention.

The carbon electrode has a larger sheet resistance value than the metal electrode, which is one of the factors that hinders the high efficiency of carbon-based PSCs. Furthermore, for MPLE-PSCs, a carbon electrode with good permeability for perovskite precursor solution is preferred, because the perovskite precursor solution has to be sufficiently transferred from the carbon surface into the scaffold layers. In previous published studies that focused on carbon electrodes of MPLE-PSCs, carbon electrodes based on ultrathin graphite and needle coke were applied and achieved *PCE* of 14.07% and 11.66%, respectively [10,11]. M. Duan, et al. applied ultrathin graphite to MPLE-PSCs instead of bulk graphite to increase the specific surface area of the carbon layer and improved hole transport performance. There are various types of graphite and carbon black that make up the carbon electrode, and the material conditions should be optimized further. Therefore, it is necessary to check out carbon materials suitable for MPLE-PSCs.

In this work, we investigated the effect of simple morphology such as the shape and size of graphite on the device using two types of graphite that do not have a large specific surface area such as ultrathin graphite [10]. MPLE-PSCs were fabricated using amorphous and pyrolytic graphite-based carbon electrodes. On the one hand using the amorphous graphite-based carbon (AGC) electrode, the *PCE* was only 5.97%. On the other hand, using pyrolytic graphite-based carbon (PGC), the *PCE* was improved to 13.45%. Our study demonstrates that the morphology and structure of the graphite of the carbon electrode affect both the sheet resistivity as well as the wettability and permeability of perovskite precursor solution which have a strong impact on the *PCE*.

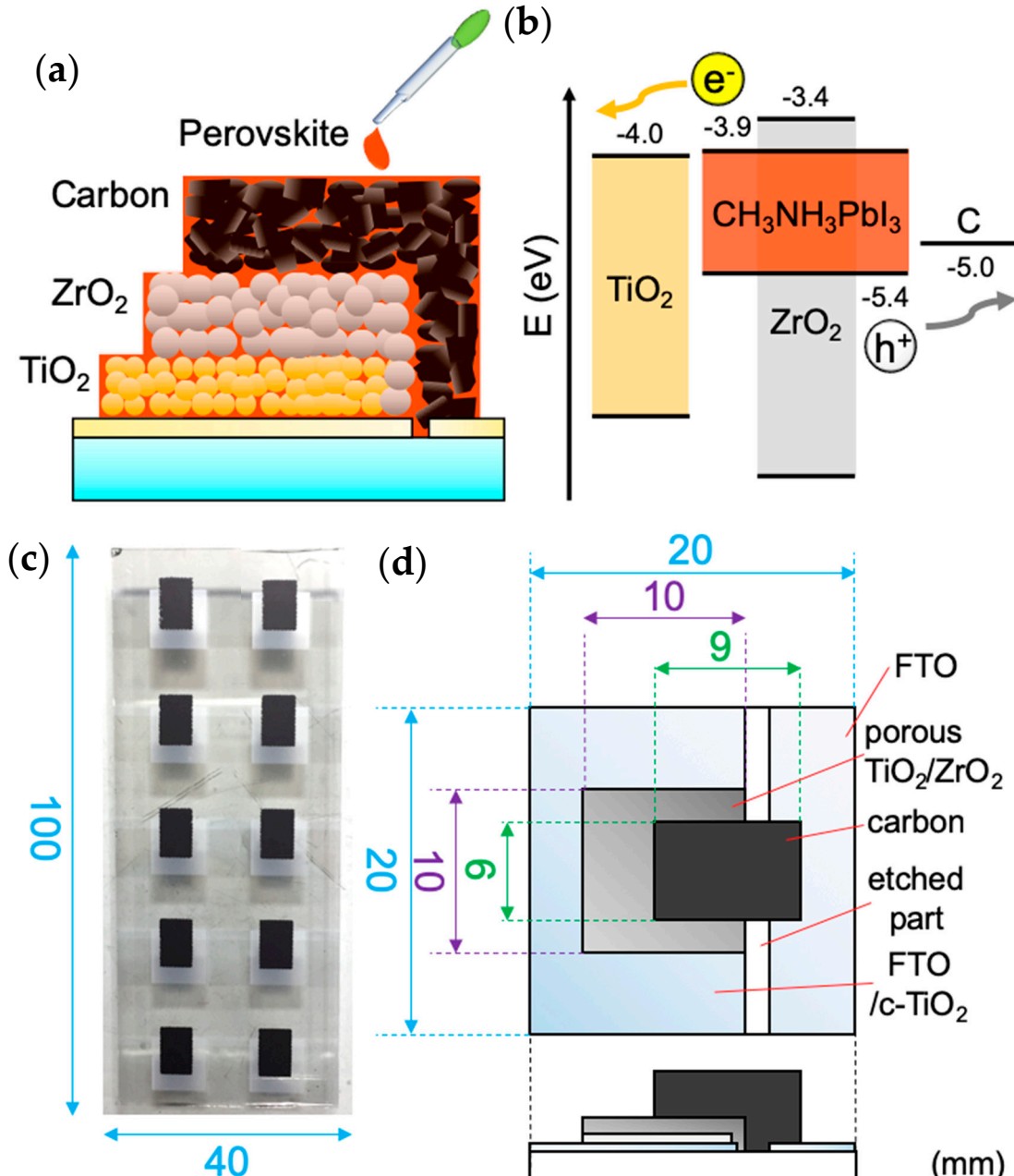

**Figure 1.** Schematic images and photograph of multiporous-layered-electrode perovskite solar cells (MPLE-PSCs). (**a**) Structure; (**b**) Energy band [4]; (**c**) The printed sheet of electrodes; (**d**) The printed multi porous electrode layers.

## 2. Experiment

### 2.1. Materials

The device employed in this work was fabricated according to procedures previously reported in the literature [4,12–14]. All the commercial materials were used as received without any purification including lead iodide (PbI$_2$, 99.99%, Tokyo Chemical Industry Co., Ltd., Tokyo, Japan); methylammonium iodide (MAI, 98.0%, Tokyo Chemical Industry Co., Ltd., Tokyo, Japan); 5-ammonium valeric acid iodide (5-AVAI, Greatcell Solar); γ-butyrolactone (GBL, for electrochemistry, Kanto Chemical Co., Inc., Tokyo, Japan); titanium diisopropoxide bis (acetylacetonate) (75 wt.% in isopropanol, Sigma-Aldrich, St. Louis, MI, USA); titanium (IV) dioxide paste (TiO$_2$, PST-30NRD, JGC Catalysts and Chemicals Ltd., Kanagawa, Japan); zirconium dioxide pastes (ZrO$_2$, Zr-Nanoxide

ZT/SP, Solaronix SA, Aubonne, Switzerland); ethanol (99.5%, Hayashi Pure Chemical Ind., Ltd., Osaka, Japan); α-terpineol (Kanto Chemical Co., Inc., Tokyo, Japan); and fluorine-doped tin oxide (FTO) glass substrate (TEC-15, Nippon Sheet Glass-Pilkington Co., Ltd., Osaka, Japan). The carbon pastes using amorphous (AT-No. 40, averaged particle size 7 μm, Oriental Industry Co. Ltd., Yamanashi, Japan) and pyrolytic graphite (PC-30, averaged particle size 30 μm, Ito graphite Co., Mie, Japan) were prepared using a modification of a previously reported procedure [13,15–18].

## 2.2. Preparation Method of Perovskite Precursor Solution

Preparation of the precursor solution was performed in a glove box filled with $N_2$ gas (the dew point was −20 to −18 °C). The 1.2 M, $(5\text{-AVA})_{0.05}(MA)_{0.95}PbI_3$ perovskite precursor solution was prepared by mixing 0.5532 g $PbI_2$, 0.1812 g MAI, and 0.0147 g 5-AVAI in 1 mL GBL, and then stirred at 60 °C and 800 rpm for 2 h.

## 2.3. Device Fabrication

The complete device fabrication process was performed under ambient air conditions (room temperature was 15 to 25 °C and the relative humidity was 30% to 50% RH). The conductive layer of the FTO glass (100 × 40 mm) (Figure 1c,d) was separated by etching using zinc powder (75~150 μm, 99.9%, Fujifilm Wako Pure Chemical Co., Osaka, Japan) and the hydrochloric acid solution diluted with distilled water (1 M, Kishida Chemical Co., Ltd., Osaka, Japan). After the etching reaction, the etched part was rubbed with a cotton swab and rinsed vigorously with water. The electrical insulation between the two parts of FTO layers was confirmed using a multimeter, and then, etched substrate was ultrasonically cleaned with a detergent solution (1%, white 7-AL, Yuai Kasei, Hyogo, Japan) and ethanol for 15 min, respectively. A blocking layer of compact $TiO_2$ (c-$TiO_2$) was deposited on patterned substrates by spray pyrolysis deposition at 500 °C on the hot plate using a 0.66 mL titanium diisopropoxide bis (acethylacetonate) solution diluted in 22.5 mL ethanol (1:34 volume ratio). The spray interval was 10 s and sprayed evenly on the entire hot plate. The 0.55 μm thick mesoporous $TiO_2$ (m-$TiO_2$) layer was screen printed using the paste prepared via mixing $TiO_2$ paste with α-terpineol in a 1:2 weight ratio, kept at room temperature for 5 min, and then dried on a hot plate at 125 °C for 10 min. The 1.7 μm thick mesoporous $ZrO_2$ (m-$ZrO_2$) spacer layer was screen printed using the $ZrO_2$ paste, kept at room temperature for 10 min, and then dried on a hot plate at 125 °C for 10 min. The $TiO_2$ and $ZrO_2$ layers were annealed at 500 °C for 1 h in electric furnace (temperature rising time was 30 min, holding time at 500 °C was 30 min). The 23 μm thick carbon-graphite layer was screen printed using AGC- or PGC-based carbon paste, kept at room temperature for 5 min, and then dried on a hot plate at 125 °C for 10 min. The carbon-graphite layer was annealed at 400 °C for 31 min (temperature rising time was 5 min, holding time at 400 °C was 26 min) in an electric furnace. Then, the substrate was cooled to room temperature and cut into single cells, following ultrasonic soldering to attach electrical contacts onto both electrodes. The area around the scaffold layer was masked with heat resistant polyimide tape so that the perovskite solution would fill in the scaffold layers. Finally, 2.0 μL of the 1.2 M $(5\text{-AVA})_{0.05}(MA)_{0.95}PbI_3$ perovskite precursor solution was supplied to the scaffold layers by drop-casting. The filled devices were kept with a glass-lid cover at room temperature for 30 min, and then heated at 50 °C on a hot plate with glass-lid cover for 30 min. Then, the cover was removed and the device was dried at 50 °C for 1 h to complete the fabrication process. The perovskite filling process was performed in ambient air (room temperature was 15 °C to 25 °C and the relative humidity was 30% to 50% RH).

## 2.4. Characterization

All the measurements were carried out under ambient conditions without any sealing. Sheet resistance measurement samples were fabricated by printing carbon paste on a glass slide masked to 1 × 1 cm square by the doctor blade method, sintering it, and then soldering both ends. Sheet resistance was measured by applying a multimeter to the solder parts at both ends of this sample.

The crystal structure of the carbon electrode was characterized by XRD (MiniFlex II, Rigaku, Austin, TX, USA). The SEM images of carbon electrode and devices were obtained using a scanning electron microscope (SEM, JSM-6510, JEOL, Akishima, Tokyo). The specific surface area was calculated from the BET (Brunnauer, Emmett, TeIler) plot, and the pore size distribution of the carbon electrodes was calculated from nitrogen ($N_2$) gas-adsorption and desorption isotherms. The wettability of the carbon electrode was evaluated by measuring the contact angle (FLOWDESIGN CAM-003) using the distilled water. The EDX element and mapping analysis of carbon electrode and devices were characterized by energy dispersive X-ray spectroscopy (EDX, TM3030, HITACHI, Tokyo, Japan). The ATR and reflectance FT-IR spectrum was analyzed by Fourier transform infrared spectroscopy (FT-IR, LUMOS, Bruker, MA, USA). After taking the photograph, the color of the active area of the device was divided into red, blue, and green values (RGB color model) using the color picker function in the Microsoft Paint software. The photocurrent density-voltage (*J-V*) curves were measured with a DC voltage current source (B2901A, Agilent, CA, USA) under a solar simulator (AM1.5G, 100 mW cm$^{-2}$) equipped with a 500 W xenon (Xe) lamp (YSS-100A, Yamashita Denso, Long Beach, CA, USA). The power of the AM1.5G solar simulator was calibrated using a reference Si photodiode (Bunkou-Keiki Co., Ltd., Tokyo, Japan). The light-irradiation area was $3 \times 3$ mm (the active area of the device was 0.36 cm$^2$ and the mask opening area was 0.09 cm$^2$). The measurement voltage ranged from −0.05 to 1.05 V with forward and reverse scan, the step was set to 0.01 V, the integration time was set to 16.7 ms, and the scan delay time was 40 ms (the measured scan rate was 110 mV s$^{-1}$). The measurements were performed five times, with an interval of 3 min between each measurement, and the device continued to be illuminated with light between measurements for the photo activation [16]. The AM1.5G light aging was performed under open-circuit condition. For the light stability test, the device was measured continuously for 10 h under simulated sunlight of AM1.5G at open circuit conditions without any encapsulation. The device was measured every 3 min. The incident photon-to-current efficiency (IPCE) spectra were measured using a 150 W Xe lamp (TSM-K1, BSO-X150) equipped with a monochromator (MHM-K1) as a monochromatic light source. Calibration with the silicon photodiode (Bunkou-Keiki Co., Ltd., Tokyo, Japan) was carried out prior to the IPCE measurements. The IPCE measurement was performed 5 times, and the interval was set to 3 min for the weak-photo activation [16]. During that interval, LED light (ELPA, Osaka, Japan, SP0T-LLC01, DC 3.5 V, 300 mA) was irradiated from the side of the device and light-soaking was performed. The current value of the sample was 20 to 30 μA, which is 0.238 to 0.357% intensity of sun simulated light (AM1.5G) at the LED light activation [16]. The distance between the device and the LED light was fixed to 10 mm. The LED light was switched off during IPCE measurement. In *J-V* and IPCE measurement, the stabilized results of the 5th measurement were used and compared. Electrochemical impedance spectroscopy (EIS) of the devices was performed on electrochemical workstations (Bio-Logic, Seyssinet-Pariset, France) in the frequency range from 1 Hz to 100 kHz at 0 V bias in dark.

## 3. Results and Discussion

The layer thickness of AGC and PGC electrodes were 23.6 ± 1.7 and 23.0 ± 1.3 μm, and the sheet resistances were 23.9 ± 0.7 and 3.6 ± 0.3 Ω/□, respectively, which was measured using the square-shaped carbon layers on glass substrates with probe metal contacts on two edges at opposite sides. Hence, the sheet resistance value of AGC is higher than that of PGC. The PGC has a very low sheet resistance value as compared with the previously reported carbon electrodes [10,11].

The XRD patterns of each carbon electrode is shown in Figure 2. A comparison of the measured XRD peak intensities of graphite (002) showed that the peak intensity of PGC was 18-times higher than that of AGC. In addition, when the peak intensities of graphite (002) of AGC and PGC were normalized, the PGC peak was sharper than that of AGC, indicating that it had high crystallinity

(Figure 2b). In addition, the AGC and PGC electrode particle sizes, calculated using the Scherrer equation (Equation (1)) shown below, were 29.7 and 66.0 nm, respectively:

$$D = \frac{k\lambda}{\beta cos\theta} \left(\beta = \frac{\pi}{180} \times \text{FWHM}\right) \tag{1}$$

where $D$ is average crystallite size (nm), $k$ is the Scherrer constant, $\lambda$ is the X-ray wavelength (here, with CuK$\alpha$, 0.15406), FWHM is the full width at half maximum of peak, and $\beta$ is the radian of FWHM. The crystal size from (Equation (1)) does not represent the actual crystallite size of graphite [19,20], due to the size limitation of (Equation (1)). In this case, simply, the variation of crystal size in the short range can be confirmed.

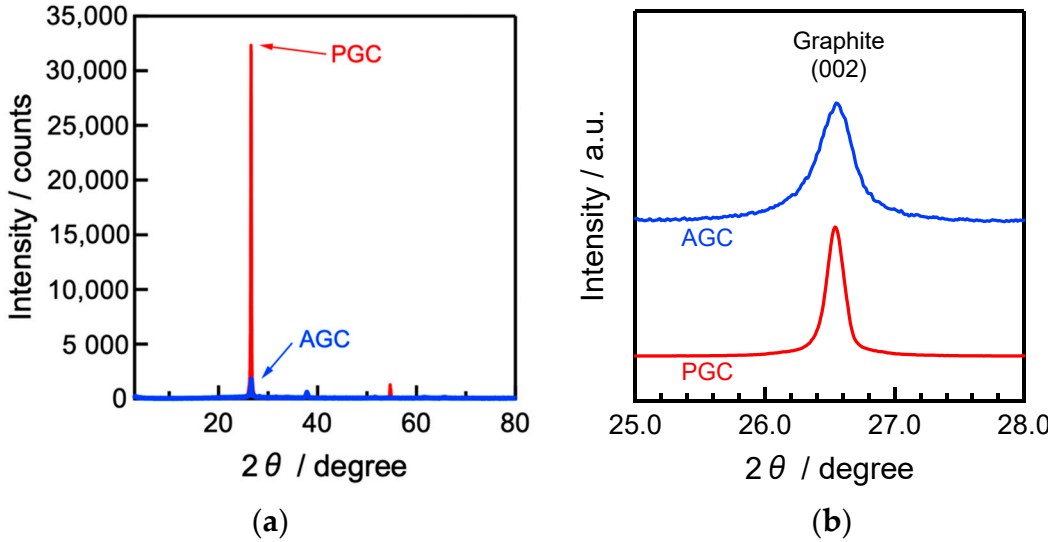

**Figure 2.** XRD patterns of amorphous graphite-based carbon (AGC) and pyrolytic graphite-based carbon (PGC) electrode. (**a**) 3–80°; (**b**) Normalized data, 25–28°.

Therefore, it is considered that PGC has higher conductivity than AGC because of the higher crystallinity of PGC than that of AGC. Figure 3 shows a SEM image of AGC and PGC electrodes. The graphite particle size was also confirmed by SEM. It was confirmed that the particle size of AGC is smaller than that of PGC (Figure 3a–d). Additionally, from the SEM cross-section image, the PGC was a larger plate-like particle (Figure 3f).

Figure 4a shows the nitrogen gas absorption and desorption isotherms of the AGC- and PGC-based carbon electrodes. Calculated by BET (Brunnauer, Emmett, Teler) method, the specific surface area value of AGC and PGC is 104.6 and 76.7 m$^2$ g$^{-1}$, respectively. Figure 4b shows the pore-size distributions of AGC and PGC electrodes and Figure 4c is the pore-volume logarithmic plots of Figure 4b. Although the distribution of pore sizes was almost the same, it was found that the number of pores was larger in AGC than PGC. In fact, AGC has a larger specific surface area than PGC, which is considered to be advantageous for the diffusion of the precursor solution. The ATR and reflectance FT-IR spectra of AGC and PGC electrodes were measured, but there was no noticeable difference between the spectra. Similarly, EDX elemental analysis on the surfaces of AGC and PGC electrodes showed almost the same elemental ratio, and no difference was observed. Therefore, the difference between AGC and PGC electrodes is mainly in morphology such as particle size and porosity.

Figure 5a,b shows optical images of contact angle measurements using distilled water on AGC and PGC electrodes. Figure 5c shows the contact-angle variation in the time course. During the same time interval (Figure 5a,b), the distilled water can be adsorbed in the AGC electrode more quickly than the PGC one. Therefore, AGC accelerates diffusion of hydrophilic solution, which may be due to the small particle size large specific surface area, and large pore volume.

Figure 6a,b shows a cross-sectional SEM image of the completed PGC-based solar cell filled with perovskite solution. The layer thicknesses of m-TiO$_2$ and m-ZrO$_2$ were 0.55 and 1.7 μm, respectively. Figure 6c,d shows the EDX mapping analysis of the cross-section in the mesoscopic layer of PSCs based on AGC and PGC electrodes, which can confirm the perovskite filling by EDX elemental mapping of the device cross-section. The same amount of perovskite precursor solution was drop cast to each of the AGC- and PGC-based devices, but there was a difference in the distribution of perovskite crystals in the m-ZrO$_2$ and m-TiO$_2$ layers. The PGC-based device clearly detected Pb and I, while the AGC-based device had lower detection intensity, indicating imperfect perovskite crystal filling. In AGC, the perovskite precursor solution that was not filled in the m-ZrO$_2$ and m-TiO$_2$ layers is considered to have accumulated in the carbon layer (the carbon layer has a thickness of ~23 μm) above the EDX mapping range (~5 μm from m-TiO$_2$ layer).

Figure 7a shows a photograph of the light-receiving side of the completed solar cells. In these MPLE-PSCs, the color of the light-receiving area can change depending on the quality of the perovskite precursor filling and its crystallization. A well-filled cell can be partially identified by its dark visual appearance, whereas poorly infiltrated cells might not only have homogeneities but also less intense absorption in the visible wavelength range, making them appear less dark. If it is imperfect, the area can be white by light scattering from the m-TiO$_2$ and m-ZrO$_2$ layers [4]. As can be seen from Figure 7a, the light-receiving area became very dark due to the effects of pore filling by perovskite through PGC. Furthermore, the color of the active area of the fabricated cells were divided into red, green, and blue values according to the RGB color model [21]. Figure 7b shows the average value and standard deviation of the colors extracted from the five parts of active areas of the cell. The smaller the RGB values, the darker the color, which shows that PGC-based devices are darker than AGC-based devices. The area using AGC was whiter than that using PGC, which we associate with inefficient penetration and filling of perovskite through PGC.

The device performance of solar cells with different carbon electrodes was measured by the photocurrent density-voltage (*J-V*) measurement using a solar simulator and the incident photon-to-current efficiency (IPCE) measurement. Figure 8a shows the *J-V* curves of the fabricated device using AGC and PGC under solar-simulated irradiation (AM1.5G). The devices performance parameters are summarized in Table 1. On the one hand, the champion device with PGC-based solar cell shows a short-circuit current ($J_{sc}$) of 21.09 mA cm$^{-2}$, an open-circuit voltage ($V_{oc}$) of 0.952 V, and a fill factor (*FF*) of 0.670, yielding a *PCE* of 13.45% in the forward scan. On the other hand, in AGC, $J_{sc}$ of 11.64 mA cm$^{-2}$, $V_{oc}$ of 0.901 V, *FF* of 0.569, yielding a *PCE* of 5.97%. In addition, with both AGC and PGC devices, the hysteresis effect on the reverse and forward scans was not noticeable. Therefore, the filling condition of perovskite did not significantly affect the hysteresis. In all the parameters, AGC showed a lower value than PGC, and especially $J_{sc}$ of AGC was as low as 11.64 mA cm$^{-2}$. This is probably due to the low amount of perovskite inside the TiO$_2$-ZrO$_2$ porous layers. Figure 8b shows the statistical results of the *J-V* measurement of AGC- and PGC-based PSC devices. The reproducibility was evaluated by the forward and reverse scan of photovoltaic characteristics using 20 PSC devices based on AGC and PGC each. The averaged $J_{sc}$, $V_{oc}$, *FF* values from the forward scan of *J-V* curves are 8.92 ± 1.26 mA cm$^{-2}$, 0.82 ± 0.04 V, 0.52 ± 0.86, and *PCE* of 3.85 ± 0.86% for AGC, and for PGC the values were 19.14 ± 0.97 mA cm$^{-2}$, 0.91 ± 0.03 V, 0.63 ± 0.03 and *PCE* of 10.14 ± 1.73%, respectively.

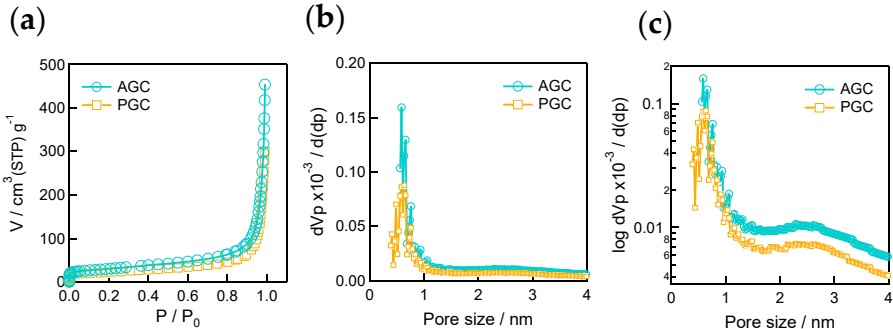

**Figure 3.** SEM image after sintering of carbon electrode using different graphite. (**a**,**c**,**e**) AGC; (**b**,**d**,**f**) PGC. (**a**–**d**) Surface view; (**c**,**d**) Enlarged view; (**e**,**f**) Cross-section view.

**Figure 4.** (**a**) Nitrogen gas adsorption and desorption isotherms of AGC and PGC electrodes; (**b**) Pore-size distributions (analyzed by adsorption and desorption isotherms) of AGC and PGC electrodes; (**c**) Pore volume log plot of Figure 4b.

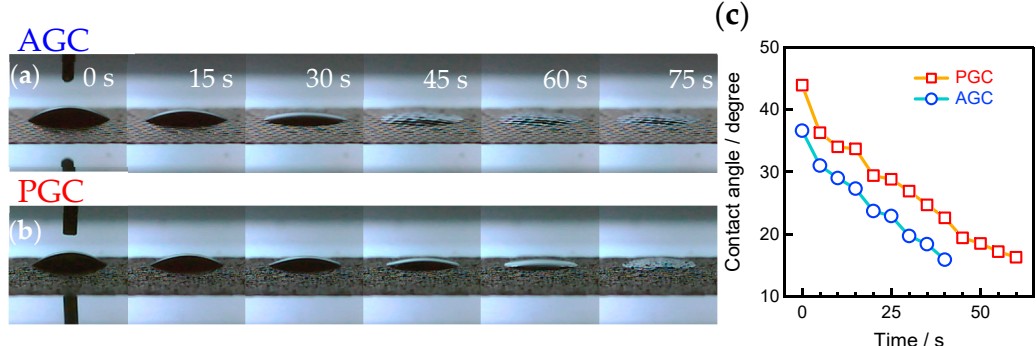

**Figure 5.** Contact angle measurement of carbon electrodes. (**a**,**b**) Optical image; (**c**) Time course of the contact angle.

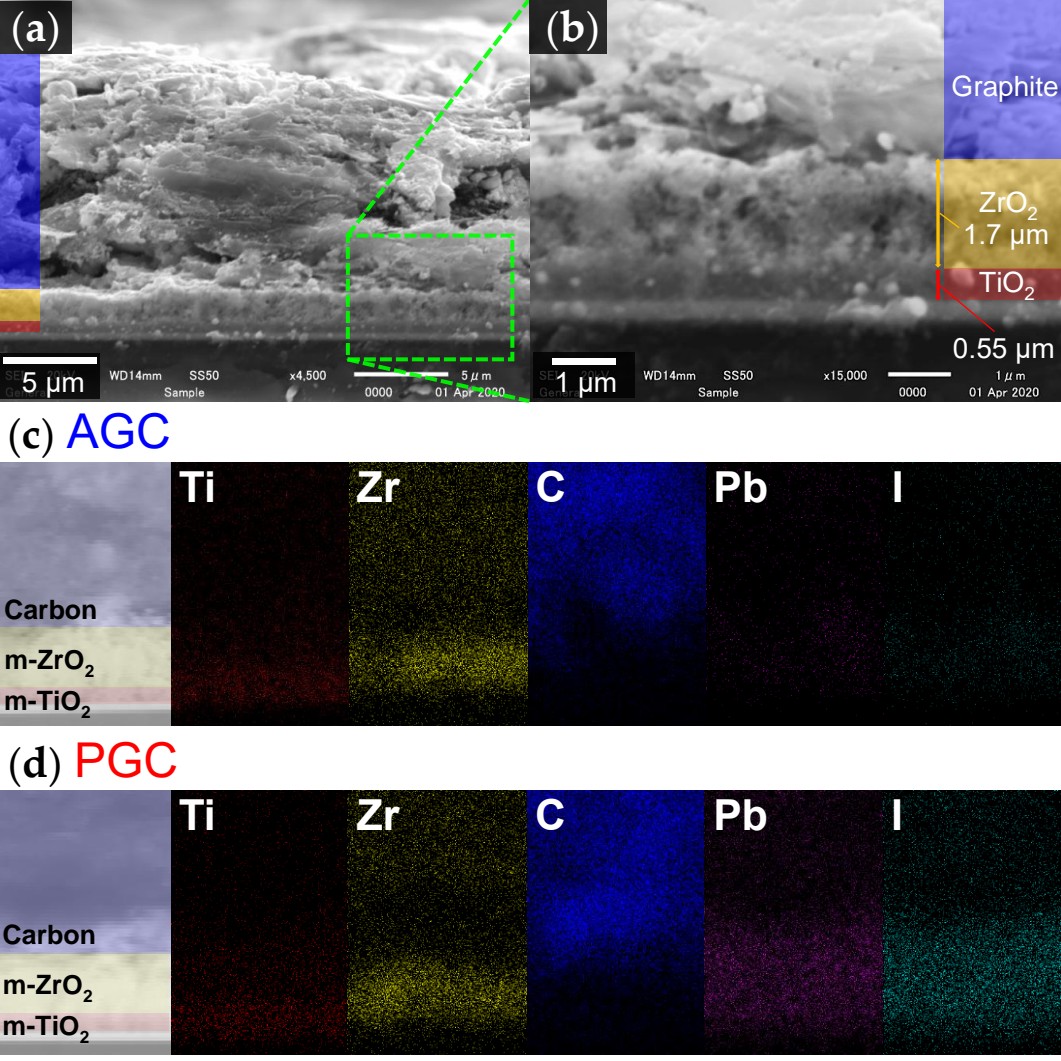

**Figure 6.** (**a**) Cross-sectional SEM image of a completed device based on PGC electrode; (**b**) Magnified image of Figure 6a; (**c**,**d**) EDX mapping analysis of a completed device based on AGC and PGC electrodes.

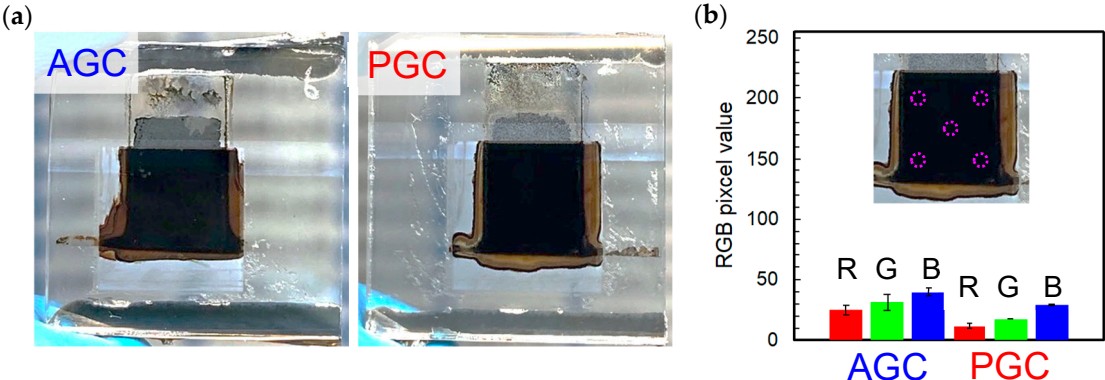

**Figure 7.** (**a**) Photographs of completed devices based on AGC and PGC electrodes; (**b**) The color of the active area sides of the AGC- and PGC-based devices. The average pixel values for red, green, and blue (RGB) in the active area are displayed with averaged value standard deviation. The inset shows the location of the extracted colors.

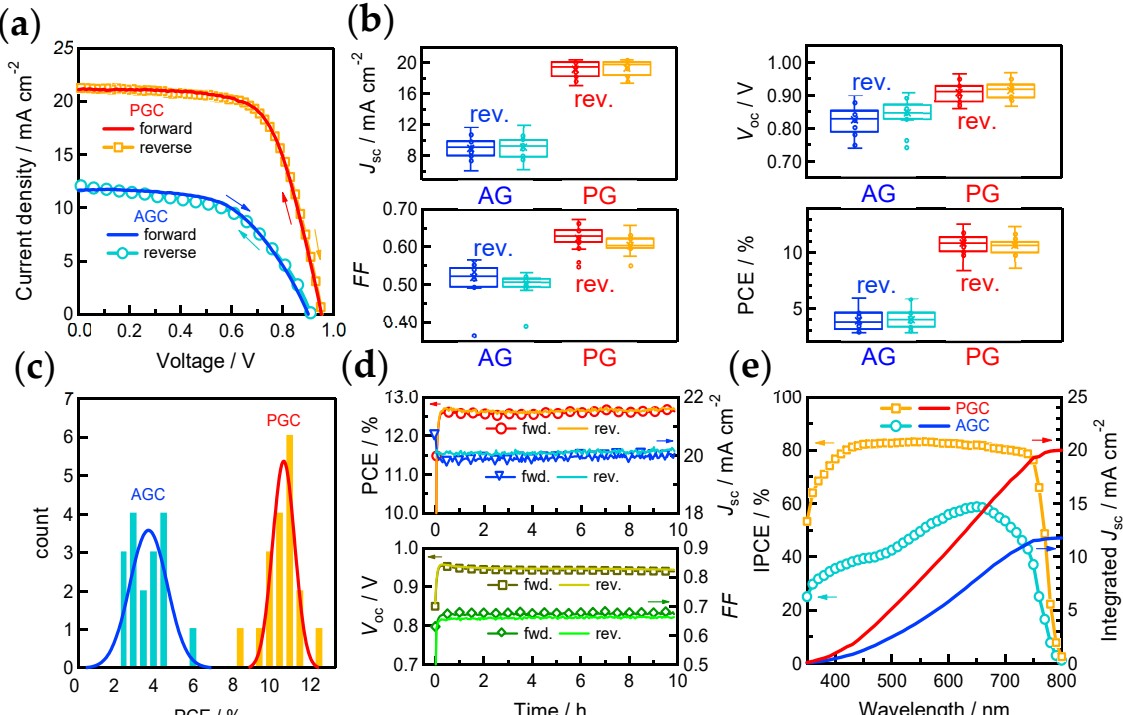

**Figure 8.** Photovoltaic performance parameters on MPLE-PSCs. (**a**) Photocurrent density-voltage (*J-V*) curve; (**b**) Box and whisker plots of each parameter for devices with AGC and PGC electrodes (forward and reverse scans are marked as fwd. and rev., respectively); (**c**) *PCE* histogram for devices with AGC and PGC electrodes; (**d**) Indoor light stability test of device with PGC electrode. The device was without any encapsulation and kept for 10 h in ambient air at RT and 50~70% RH. During the test, the device was continuously illuminated with simulated sunlight of AM1.5G. The device was measured every 3 min; (**e**) Incident photon-to-current efficiency (IPCE) performance of champion devices based on AGC and PGC electrodes.

**Table 1.** Performance parameters of champion devices based on AGC and PGC electrodes.

| Carbon Electrode | Scan | $J_{sc}$/mA cm$^{-2}$ | $V_{oc}$/V | FF | PCE/% |
|---|---|---|---|---|---|
| AGC | forward | 11.64 | 0.901 | 0.569 | 5.97 |
| | reverse | 12.05 | 0.912 | 0.527 | 5.81 |
| PGC | forward | 21.09 | 0.952 | 0.670 | 13.45 |
| | reverse | 21.31 | 0.955 | 0.658 | 13.38 |

The *PCE* histogram for each of the 20 devices with AGC and PGC electrodes are shown in Figure 8c. On the one hand, AGC had lower *PCE* value and poor reproducibility due to incomplete filling of perovskites. On the other hand, PGC showed higher *PCE* due to the complete filling of perovskite, and the higher reproducibility was also good. The low *PCE* of AGC-based PSCs is mainly due to the low $J_{sc}$ value. The PSCs using PGC showed high reproducibility in both forward and reverse scans, whereas using AGC, the $J_{sc}$ value showed large fluctuation (Figure 8c). Regarding the fluctuation of the $J_{sc}$ value of AGC-based PSCs, it is suggested that the filling of the perovskite in the porous layer is inhomogeneous in each cell.

Figure 8d shows the results of the light-stability test of PGC-based PSCs without any encapsulation under continuous simulated sunlight (AM1.5G) at open circuit conditions. During the 10 h of continuous light irradiation, $J_{sc}$ increased slightly and $V_{oc}$ decreased slightly, but the performance remained at 100% of the initial value and no significant change was observed. PGC-based PSCs showed high stability without degradation of *PCE* during 10 h of continuous light irradiation due to the complete filling of the perovskite within the scaffold layer.

Figure 8e shows the result of incident photon-to-current efficiency (IPCE) measurement of PSCs based on AGC and PGC and the integrated $J_{sc}$ value calculated from the IPCE spectra. PGC showed high IPCE values in all areas from short wavelength to long wavelength, with an integrated $J_{sc}$ value of 20.00 mA cm$^{-2}$. Whereas, in AGC, showed lower efficiency in the short wavelength area as compared to the long wavelength area, with an integrated $J_{sc}$ value of 11.76 mA cm$^{-2}$. We suppose that the low IPCE using AGC in the short wavelength region is caused by the insufficient filling of the perovskite into the porous layer as supported by EDX-analysis (Figure 6), and by blue light scattering effect of the m-TiO$_2$ and m-ZrO$_2$ scaffold layer nanoparticles [18]. Consequently, the short wavelength light does not reach the perovskite crystal.

These results lead us to the model proposed in Figure 9 to explain the permeation of the perovskite solution into AGC- and PGC-based electrode stacks. We suggest that the perovskite precursor solution dropped on AGC, on the one hand, diffused mainly into the carbon electrode and did not reach the underlying m-TiO$_2$ and m-ZrO$_2$ layers sufficiently. In PGC, the wettability of the graphite layer is slower, but the perovskite precursor solution infiltrates the m-ZrO$_2$-TiO$_2$ layers more densely. Therefore, it should be noted that, in spite of the good wettability of the carbon electrode, the permeability to the lower layer is poor. The difference in the permeability by the component of the carbon electrode is considered to be due to the shape and particle size of graphite. Due to the appropriate particle size of graphite and the formulation of the precursor carbon paste, it was considered that the perovskite precursor solution successfully penetrated the underlayer in the PGC device. In AGC, the precursor solution spread mainly in the carbon electrode, but, in PGC, the proper carbon electrode structure improved the permeability. As a result, on the one hand, the PGC device showed high performance due to the complete penetration and filling of the precursor solution. Conversely, in AGC, the wettability was good, but the precursor spread mainly in the carbon electrode, so the performance was low. In order to fabricate the optimum carbon electrode for MPLE-PSCs, the ratio of graphite powder, carbon black, solvent, and binder, which are the components of the carbon paste, needs to be optimized. In addition, it is necessary to optimize the amount of perovskite precursor solution introduced into the scaffold layers, which depends on the porosity of the carbon electrodes. If the precursor solution is clogged inside the carbon electrode, filling the perovskite crystal into the m-ZrO$_2$-TiO$_2$ layer further is very difficult.

Figure 10 shows the Nyquist plot by electrochemical impedance spectroscopy (EIS) at 0 V bias in the dark of AGC- and PGC-based PSCs. The measurement results were fitted using the equivalent circuit of the insert in Figure 10a, and detailed fitting data for the EIS are shown in Table 2. The $R_s$, which represents the series resistance of the solar cell, was 22.0 Ω for AGC, while it was as low as 15.4 Ω for PGC (Figure 10b). The lower $R_s$ of this PGC-based PSCs is provided by the low sheet resistance of PGC electrode. The $R_p$, which indicates the parallel resistance of the solar cell, was $2.9 \times 10^5$ and $6 \times 10^5$ Ω for AGC and PGC, respectively. The lower $R_p$ value of AGC may be attributed to the leakage current caused by an imperfect diode in insufficient perovskite-filled layers. In both *J-V* and EIS measurements, $R_s$ was lower in PGC than in AGC. Similarly, the values of $R_{sh}$ of *J-V* and $R_p$ of EIS were also higher in PGC than in AGC.

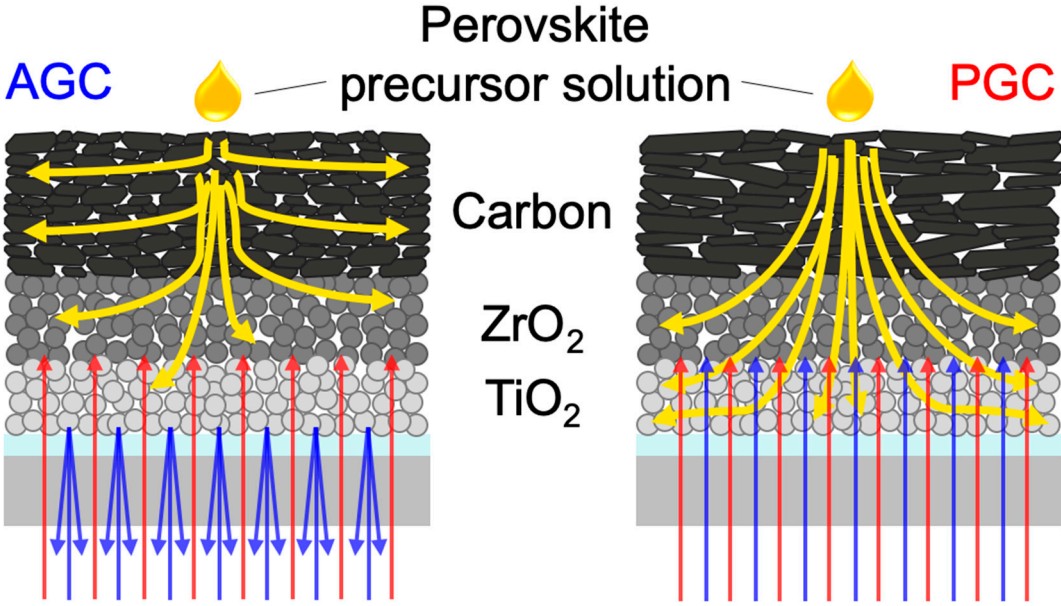

**Figure 9.** Schematic images of the different permeability of perovskite precursor solutions of the devices based on AGC and PGC electrodes (Yellow arrow, permeation direction of the perovskite precursor solution; red arrow, long wavelength light; and blue arrow, short wavelength light).

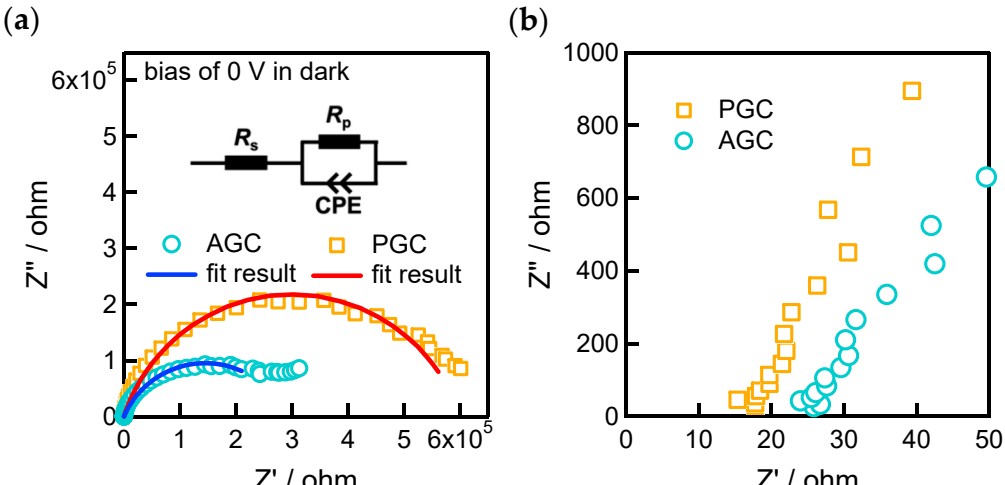

**Figure 10.** (**a**) Nyquist plots of the device based on AGC and PGC electrodes at bias of 0 V in dark. The inset shows equivalent circuit used for fitting; (**b**) Magnified image of Figure 8a.

**Table 2.** Resistance parameters obtained from *J-V* and electrochemical impedance spectroscopy (EIS) measurement of devices based on AGC and PGC electrodes. The bias at EIS was 0 V in the dark. (Series resistance of the solar cell (*R*$_s$ and *R*$_{sh}$) of *J-V* measurement are the average values of forward and reverse scans.)

| Carbon Electrode | From *J-V* Measurement | | From EIS Measurement | | | |
|---|---|---|---|---|---|---|
| | $R_s/\Omega$ | $R_{sh}/\Omega$ | $R_s/\Omega$ | $R_p/\Omega$ | *CPE-T* [F] | *CPE-P* |
| AGC | 15.7 | 709 | 22.0 | $2.9 \times 10^5$ | $4.5 \times 10^{-7}$ | 0.7 |
| PGC | 8.0 | 2038 | 15.4 | $6 \times 10^5$ | $4.5 \times 10^{-8}$ | 0.8 |

## 4. Conclusions

In summary, amorphous graphite-based carbon (AGC) and pyrolytic graphite-based carbon (PGC) electrodes were applied to the back-contact electrode of fully printable carbon-based multiporous-layered-electrode perovskite solar cells (MPLE-PSCs). AGC showed low device performance despite the good wettability. Conversely, PGC showed high device performance despite the poor wettability. The low $J_{sc}$ value and low IPCE efficiency in the short wavelength region of the AGC indicated incomplete filling of the perovskite precursor solution into the porous $TiO_2$-$ZrO_2$ layers. In the AGC electrode, the precursor solution diffused and spread into the carbon electrode but did it not reach the porous $TiO_2$-$ZrO_2$ layers sufficiently. In contrast, the PGC-based perovskite solar cells (PSCs) allowed the precursor solution to penetrate evenly into the porous $TiO_2$-$ZrO_2$ layers. As a result, the PSCs based on PGC achieved a higher performance as compared with the AGC, presenting the improvements of *PCE* from 5.97% to 13.45% efficiency and of $J_{sc}$ from 11.64 mA cm$^{-2}$ to 21.09 mA cm$^{-2}$. This study demonstrates that the graphite material is crucial to achieving high-quality penetration of the precursor solution. We identified a complex interplay between both wettability and permeability, highlighting that both properties need to be evaluated for future device optimization.

**Author Contributions:** R.T. conceived and designed the experiments, performed the experiments, and wrote the paper; D.B., D.M., L.W., E.K., R.F., S.M., and A.H. contributed the discussions; Y.M. performed N$_2$ gas adsorption and desorption isotherms measurement; S.I. managed this work comprehensively. All authors have read and agreed to the published version of the manuscript.

**Funding:** This work was supported by JST, EIG CONCERT-Japan, Strategic International Collaborative Research Program (SICORP), the project name of "Printable Fully Inorganic Porous Metal Oxide Based Perovskite Solar Cells: Defining Charge Selective Oxides for High-efficient and Low-cost Device Structure (PROPER)". <JAXA>. Dmitry Bogachuk acknowledges the scholarship support of the German Federal Environmental Foundation (DBU).

**Conflicts of Interest:** The authors declare no conflict of interest.

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
