# Peer review of "Function of Porous Carbon Electrode during the Fabrication of Multiporous-Layered-Electrode Perovskite Solar Cells"

_photonics, doi:10.3390/photonics7040133_

Round 1

Reviewer 1 Report

This manuscript reports the use of the graphite-based electrode for fully-printed perovskite solar cells. The authors compared the devices with amorphous graphite (AGC) and pyrolytic graphite (PGC), and the PGC-based devices exhibit much better photovoltaic performances, especially increased IPCE and Jsc. The authors performed various experiments for finding out the origin of the enhanced performances and for supporting their claims. The reviewer finds no critical issue from the results and discussions in the manuscript, so this manuscript can be accepted after addressing the following minor comments/suggestions.

  1. The authors used the Scherrer equation for calculating the particle size of the graphite, but this equation may not be used for analyzing highly crystalline materials with large crystallite size. Given the very sharp X-ray diffraction peak of PGC, its actual crystallite size might be much larger (especially in the planar direction) than the calculated value of 66.0 nm. The authors need to double-check the particle size using other tools like electron microscopy, or refer to the cited references (refs. 15-18).
  2. Regarding the infiltration model shown in Fig. 9, it would be good if the authors can provide more explanation about the reason of better permeability of PGC.
  3. Some typos and grammatical error are seen from the manuscript (e.g. unit of sheet resistance in line 167), so the authors are recommended to go through the manuscript and carefully edit it.

Author Response

Response to reviews

First of all, we would like to thank the reviewers very much for your positive comments. We have revised our manuscript as reviewer’s suggestions sincerely. The revised points in the submitted text were changed to yellow. The answers to the comments of Reviewer 1 are shown as below:

<Reviewer 1>

Comments to the Author

Comment 1.

The authors used the Scherrer equation for calculating the particle size of the graphite, but this equation may not be used for analyzing highly crystalline materials with large crystallite size. Given the very sharp X-ray diffraction peak of PGC, its actual crystallite size might be much larger (especially in the planar direction) than the calculated value of 66.0 nm. The authors need to double-check the particle size using other tools like electron microscopy, or refer to the cited references (refs. 15-18).

Reply to Referee 1 of comment 1: Thank you very much for the comments and suggestions. We have added graphite particle size information disclosed by the manufacturer. And we added an enlarged SEM surface images (Figure 3c and 3d) of the graphite electrode and revised the XRD and SEM text as follows: And we added references about graphite particle size and XRD (Scheller formulas).

(l. 97 to 99, p. 3)

The carbon pastes using amorphous (AT-No.40, averaged particle size: 7 μm, Oriental Industry Co. Ltd.) and pyrolytic graphite (PC-30, averaged particle size: 30 μm, Ito graphite Co.) were prepared

(l. 185, p. 4 to l. 196, p. 5)

And, the AGC and PGC electrode particle sizes calculated using the Scherrer equation (eq. 1) shown below were 29.7 and 66.0 nm, respectively.

(see the equation in the attached file, please)

(1)

(D: Average crystallite size (nm), k: Scherrer constant, λ: X-ray wavelength (Here, with CuKα, 0.15406), FWHM: Full width at half maximum of peak, β: is the radian of FWHM). The crystal size from (eq. 1) doesn’t represent the actual crystallite size of graphite [12, 13], due to the size limitation of (eq. 1). Simply, the variation of crystal size in the short range can be confirmed.

Therefore, it is considered that PGC has higher conductivity than AGC because of the higher crystallinity of PGC than that of AGC. Figure 3 shows a SEM image of AGC and PGC electrodes. The graphite particle size was also confirmed by SEM. It was confirmed that the particle size of AGC is smaller than that of PGC (Figure 3a, 3b, 3c and 3d). Also, from the SEM cross-section image, the PGC was a larger plate-like particle (Figure 3f).

(References)

  1. Li, Z.-Q.; Lu, C.-J.; Xia, Z.-P.; Zhou, Y.; Luo, Z. X-ray diffraction patterns of graphite and turbostratic carbon. Carbon 2007, 45, 1686-1695.
  2. Ito, S.; Takahashi, K.; Fabrication of Monolithic Dye-Sensitized Solar Cell Using Ionic Liquid Electrolyte. J. Photoenergy 2012, 915352.

Comment 2.

Regarding the infiltration model shown in Fig. 9, it would be good if the authors can provide more explanation about the reason of better permeability of PGC.

Reply to Referee 1 of comment 2: Thank you for the fruitful comments. About the model in Fig.9, we added the following text and changed the structure of the text.

(l. 284 to 293, p. 6)

Due to the appropriate particle size of graphite and the formulation of the precursor carbon paste, it was considered that the perovskite precursor solution successfully penetrated the underlayer in the PGC device. In AGC, the precursor solution spread mainly in the carbon electrode, but in PGC, the proper carbon electrode structure improved the permeability. As a result, the PGC device showed high performance due to the complete penetration and filling of the precursor solution. On the other hand, in AGC, the wettability was good, but the precursor spread mainly in the carbon electrode, so the performance was low. In order to fabricate the optimum carbon electrode for MPLE-PSCs, the ratio of graphite powder, carbon black, solvent, and binder, which are the components of the carbon paste, needs to be optimized.

Comment 3.

 Some typos and grammatical error are seen from the manuscript (e.g. unit of sheet resistance in line 167), so the authors are recommended to go through the manuscript and carefully edit it.

Reply to Referee 1 of comment 3: We are thankful to referee for the comments. We have carefully reviewed the text and amended the following:

(l. 22 to 25, p. 1)        Jsc, open-circuit photovoltage (Voc), fill factors (FF) and PCE

(l. 35, p. 1)                  photoelectric conversion efficiency (PCE) was

(l. 41, p. 1)                  (e.g., spiro-OMeTAD) and metal counter electrodes (e.g., Au or Ag)

(l. 43, p. 1)                  in 2013, Ku, et al. (H. Han’s group,

(l. 60, p. 2)                  high-performance devices.

(l. 81, p. 2)                  pyrolytic graphite-based

(l. 152, p. 4)                voltage (J-V) curves were

(l. 158, p. 4)                and the scan delay time of 40 ms

(l. 178, p. 4)                3.6±0.3 Ω/â–¡, respectively.

(l. 281, p. 6)                infiltrates the m-ZrO2-TiO2

(l. 296, p. 6)                infiltrates the m-ZrO2-TiO2

(l. 309, p. 7)                to the back-contact electrode

(l. 324, p. 7)                Y.M. performed N2 gas-adsorption

Reviewer 2 Report

Dear editor,

This paper study the Function of Porous Carbon Electrode on Multi-Porous-Layered-Electrode Perovskite Solar Cells. They specifically studied the difference between AGC and PGC electrodes in perovskite solar cells. This result has a significant contribution in this field. Therefore, this paper should be accepted after answering some questions as below.

  1. From contact angle measurement, the AGC electrode has lower CA than PGC electrode. The perovskite precursor should penetrate into the device more easily. Why it not penetrated into porous TiO2-ZrO2 layer and had a lower efficiency?
  2. Authors say that The lower JSC of AGC based PSC is due to the poor perovskite filling into the porous TiO2-ZrO2 layers. However, when prepared AGC based and PGC based PSCs, authors should drop the same amount of precursor onto device for. If it is the poor perovskite precursor filling into the porous TiO2-ZrO2 layers, it should be accumulated in some places such as in carbon electrode (authors also have mentioned in page 6). We do not see the accumulation of perovskite by the cross sectional EDX mapping analysis (Fig.6 c and d). Why?
  3. As author sat that it has poor perovskite filling in AGC based PSC. Why the hysteresis behavior still show very slight?

Author Response

Response to reviews

First of all, we would like to thank the reviewers very much for your positive comments. We have revised our manuscript as reviewer’s suggestions sincerely. The revised points in the submitted text were changed to yellow. The answers to the comments of reviewer 2 are shown as below:

<Reviewer 2>

Comments to the Author 

Comment 1.

From contact angle measurement, the AGC electrode has lower CA than PGC electrode. The perovskite precursor should penetrate into the device more easily. Why it not penetrated into porous TiO2-ZrO2 layer and had a lower efficiency?

Reply to Referee 2 of comment 1: Thank you very much for the comments. We have added a description for the question below:

(l. 287 to 291, p. 6)

In AGC, the precursor solution spread mainly in the carbon electrode, but in PGC, the proper carbon electrode structure improved the permeability. As a result, the PGC device showed high performance due to the complete penetration and filling of the precursor solution. On the other hand, in AGC, the wettability was good, but the precursor spread mainly in the carbon electrode, so the performance was low.

Comment 2.

Authors say that the lower JSC of AGC based PSC is due to the poor perovskite filling into the porous TiO2-ZrO2 layers. However, when prepared AGC based and PGC based PSCs, authors should drop the same amount of precursor onto device for. If it is the poor perovskite precursor filling into the porous TiO2-ZrO2 layers, it should be accumulated in some places such as in carbon electrode (authors also have mentioned in page 6). We do not see the accumulation of perovskite by the cross sectional EDX mapping analysis (Fig.6 c and d). Why?

Reply to Referee 2 of comment 2: We are thankful to referee for the comments. We added the explanation to the question in the text as follows:

(l. 217 to 223, p. 5)

The same amount of perovskite precursor solution was drop cast to each of the AGC and PGC-based devices, but there was a difference in the distribution of perovskite crystals in the m-ZrO2 and m-TiO2 layers. The PGC-based device clearly detected Pb and I, while the AGC-based device had lower detection intensity, indicating imperfect perovskite crystal filling. In AGC, the perovskite precursor solution that was not filled in the m-ZrO2 and m-TiO2 layers is considered to have accumulated in the carbon layer (the carbon layer has a thickness of ~ 23 μm) above the EDX mapping range (~5 μm from m-TiO2 layer).

Comment 3.

As author sat that it has poor perovskite filling in AGC based PSC. Why the hysteresis behavior still show very slight?

Reply to Referee 2 of comment 3: Thank you very much for the comments. In MPLE-PSCs, the hysteresis of reverse and forward scans of J-V measurement can be adjusted by changing the scan delay time. We are currently investigating the relationship between scan delay time and hysteresis in J-V measurement of MPLE-PSCs and are summarizing it now, so we would like to discuss it in detail in the next paper. In addition, the following text about the hysteresis of J-V measurement was added in the text.

(l. 244, p. 5, to l. 246, p. 6)

yielding a PCE of 5.97%. Also, with both AGC and PGC devices, the hysteresis effect on the reverse and forward scans was not noticeable. Therefore, the filling condition of perovskite did not significantly affect the hysteresis. In all the parameters,

Reviewer 3 Report

In the work entitled “Function of Porous Carbon Electrode during the Fabrication of Multi-Porous-Layered-Electrode Perovskite Solar Cells” the authors demonstrated the effect of sheet conductivity and perovskite infiltration in carbon electrodes applied for fully-printable carbon based multi-porous-layered-electrode perovskite solar cells. As a general comment the works is clear and the main purpose is well identified but, in order to be accepted as publication in Photonics, minor revisions are required and are following detailed:

  • In the introduction authors mentioned a previous work in which carbon electrode based on ultrathin graphite were applied in PSCs achieving PCE of 14.07% (Carbon 2017). Authors should mention what is the main difference between the graphite used in the work of M. Duan and co-workers (Carbon 2017) and the graphite proposed in their work. What are the main advantages? Why should be convenient the use of the approach proposed by the authors for fabricating carbone-based PSCs?
  • In the experimental section the authors declared a single sintering process at 500°C for both mesoporous TiO2 (m-TiO2) and mesoporous ZrO2 (m-ZrO2) layers. However as alternative procedure two sintering process (one dedicated to each mesoporous layer) could be applied. How could this impact over the final device performance? Please, specify why the author chose the first route to fabricate the PSC negative electrode and mention the alternative procedure in the introduction part.
  • On the experimental section authors declared that sheet resistance of printed carbon paste was measured by applying a multimeter to the solder parts at both ends of this sample. Could they provide more precise measurements by employing a 4-probes equipment?
  • In the “results and discussion” section the author reported the following statement: “well-filled cell can be partially identified by its dark visual appearance, whereas poorly infiltrated cells might not only have homogeneities but also less intense absorption in the visible wavelength range, making them appear less dark. If it is imperfect, the area can be white by light-scattering form the m-TiO2 and m-ZrO2 As can be seen from figure 7, the light-receiving area became very dark due to the effects of pore filling by perovskite through PGC.” Could the authors provide some systematic spatially resolved characterizations of this aspect? (For example, fluorescence map or LBIC map over of active area.)

Author Response

Response to reviews

First of all, we would like to thank the reviewers very much for your positive comments. We have revised our manuscript as reviewer’s suggestions sincerely. The revised points in the submitted text were changed to yellow. The answers to the comments of reviewer 3 are shown as below:

<Reviewer 3>

Comments to the Author

Comment 1.

In the introduction authors mentioned a previous work in which carbon electrode based on ultrathin graphite were applied in PSCs achieving PCE of 14.07% (Carbon 2017). Authors should mention what is the main difference between the graphite used in the work of M. Duan and co-workers (Carbon 2017) and the graphite proposed in their work. What are the main advantages? Why should be convenient the use of the approach proposed by the authors for fabricating carbon-based PSCs?

Reply to Referee 3 of comment 1: Thank you very much for the comments. We have added the text corresponding to this question in the manuscript as follows:

(l. 73 to 80, p. 2)

 In previous published studies focusing on carbon electrodes of MPLE-PSCs, carbon electrodes based on ultrathin graphite and needle coke were applied and achieved PCE of 14.07% and 11.66%, respectively [10,11]. M. Duan, et al. applied ultra-thin graphite to MPLE-PSCs instead of bulk graphite to increase the specific surface area of the carbon layer and improve hole transport performance. There are various types of graphite and carbon black that make up the carbon electrode, and the material conditions should be optimized further. Therefore, it is necessary to check out carbon materials suitable for MPLE-PSCs.

In this work, we investigated the effect of simple morphology such as the shape and size of graphite on the device using two types of graphite that do not have a large specific surface area like ultrathin graphite [10]. MPLE-PSCs were fabricated using amorphous and pyrolytic graphite-based carbon electrodes.

Comment 2.

In the experimental section the authors declared a single sintering process at 500°C for both mesoporous TiO2 (m-TiO2) and mesoporous ZrO2 (m-ZrO2) layers. However as alternative procedure two sintering process (one dedicated to each mesoporous layer) could be applied. How could this impact over the final device performance? Please, specify why the author chose the first route to fabricate the PSC negative electrode and mention the alternative procedure in the introduction part.

Reply to Referee 3 of comment 2: We are thankful to referee for the comments. As you say, there are ways to sinter m-TiO2 and m-ZrO2 simultaneously or separately. We are currently considering these two methods and would like to discuss them in detail in our next paper on layered electrodes optimization. And We added the following sentence to this question.

(l. 48 to 50, p. 2)

 All layers can be deposited by screen printing processes. In general, the m-TiO2 layer and m-ZrO2 layer are sintered at 500 °C, and the carbon layer is sintered at 400 °C. The m-TiO2 and m-ZrO2 layer are sintered simultaneously or separately. In this work, the MPLE-PSCs were fabricated by simultaneous sintering, which is a simpler fabricating process. As the final processing step,

Comment 3.

On the experimental section authors declared that sheet resistance of printed carbon paste was measured by applying a multimeter to the solder parts at both ends of this sample. Could they provide more precise measurements by employing a 4-probes equipment?

 Reply to Referee 3 of comment 3: Thank you very much for the comments. First, when the sheet resistance of the printed carbon electrode was measured using multimeter, the AGC and PGC were 23.9 ± 0.7 and 3.6 ± 0.3 Ω/â–¡, respectively. Then, we tried to measure the resistance value of the deposited carbon electrode with a 4-pin probe instrument, but the needle of the probe broke through the carbon and we could not measure it well. Therefore, we are very sorry, but we cannot provide the resistance value by the 4-probe device. However, the resistance value of the electrode is supplemented by the sheet resistance measurement, Rs value of J-V measurement, and Rs value of EIS. And We added a sentence corresponding to this question.

(l. 178 to 179, p. 4)

 resistances were 23.9±0.7 and 3.6±0.3 Ω/â–¡, respectively, which was measured using the square shaped carbon layers on glass substrates with probe metal contacts on two edges at opposite sides.

Comment 4.

In the “results and discussion” section the author reported the following statement: “well-filled cell can be partially identified by its dark visual appearance, whereas poorly infiltrated cells might not only have homogeneities but also less intense absorption in the visible wavelength range, making them appear less dark. If it is imperfect, the area can be white by light-scattering form the m-TiO2and m-ZrO2 As can be seen from figure 7, the light-receiving area became very dark due to the effects of pore filling by perovskite through PGC.” Could the authors provide some systematic spatially resolved characterizations of this aspect? (For example, fluorescence map or LBIC map over of active area.)

Reply to Referee 3 of comment 4: Thank you very much for the comments. We are very sorry that we do not have the equipment to earn the fluorescence map or LBIC map over of active area, so we could not provide these data. However, we were able to evaluate the blackness of AGC and PGC-based devices by using the RGB color model. The explanation for this comment has been added to the Manuscript as follows. And we added one reference about the RGB color model.

( l. 149 to 151, p. 4)

 2.4. Characterization

by Fourier transform infrared spectroscopy (FT-IR, LUMOS, Bruker). After taking the photograph, the color of the active area of the device was divided into red, blue, and green values (RGB color model) using the color picker function in the Microsoft Paint software. The photocurrent density-

(l. 231 to 235, p. 5) 

    1. Results and Discussion

perovskite through PGC. Furthermore, the color of the active area of the fabricated cells were divided into red, green, and blue values according to the RGB color model [21]. Figure 7b shows the average value and standard deviation of the colors extracted from the five parts of active areas of the cell. The smaller the RGB values, the darker the color, which shows that PGC-based devices are darker than AGC-based devices. The area using AGC was whiter than that using PGC,

References

  1. Hashmi, S.-G.; Tiihonen, Armi.; Martineau, David.; Ozkan, Merve.; Vivo, Paola.; Kaunisto, Kimmo.; Ulla, Vainio.; Zakeeruddin, S.-M.; Grätzel, M. Long term stability of air processed inkjet infiltrated carbon-based printed perovskite solar cells under intense ultra-violet light soaking. J. Mater. Chem. A 2017, 5, 4797-4802.